# Highly Pathogenic Avian Influenza Viruses (HPAIV) Associated with Major Southern Elephant Seal Decline at South Georgia

Connor. C. G. Bamford [1,5] ✉, Nathan Fenney[1,5], Jamie Coleman [1,5], Cameron Fox-Clarke[1], John Dickens[1], Mike Fedak [2], Peter Fretwell [1], Luis Hückstädt [3] & Phil Hollyman [1,4]

The emergence of highly pathogenic avian influenza viruses (HPAIV) has caused widespread mortality wildlife globally. In 2023, mass mortalities of southern elephant seals *Mirounga leonina* were observed in South America, and the virus subsequently reached the sub-Antarctic, affecting multiple species. The remoteness of these islands has limited assessment of its true impact. Here we present evidence of HPAIV's effect on the number of breeding females at the world's largest southern elephant seal population at South Georgia. Following the virus' arrival in 2023, we recorded a 47% (SD = 14.2%) decline in the number of breeding females at the three largest breeding colony beaches in 2024 compared to 2022. The apparent loss of nearly half the breeding female population has serious implications for recruitment and future stability of the population. These findings highlight the urgent need for continued, intensive monitoring to track the long-term effects on this species.

Southern elephant seals *Mirounga leonina* are the largest of the pinnipeds, and are a major predator with a circumpolar distribution in the Southern Hemisphere[1]. Southern elephant seals breed annually and come ashore on sub-Antarctic islands[2] at the end of the Austral winter, where they form dense colonies comprised of competitive harems on beaches[3–5]. Large males haul out first, towards the end of September and early October, followed by pregnant females, which give birth ~3–5 days post-arrival[6,7]. Pups are then weaned approximately 22–23 days post-partum, with females coming into oestrus several days prior to weaning[6,8,9]. The number of female arrivals and departures follows a Gaussian (normal) distribution, where numbers are typically 50% of the maximum two to three weeks either side of the breeding peak[3,6,7,10], which, at South Georgia, has historically fallen during the final week of October[3,11,12].

Four genetically distinct populations have been identified within the Southern Hemisphere: the Peninsula Valdés population in Argentina; (ii) the South Georgia population, which includes the South Sandwich and Falkland Islands, in the South Atlantic; the Macquarie population in the South Pacific; and finally the Heard and Kerguelen populations, which includes the Crozet and Prince Edward archipelagos, in the south Indian Ocean[13,14]. These four broad areas comprise the principal breeding locations for this species in the Southern Ocean.

Population trajectories are varied throughout the Southern Ocean. The Peninsula Valdés population in Argentina has been growing at between 1 and 3.4% annually for the last five decades[15]. At its last census in 1995, the South Georgia population was deemed to be stable and accounted for ~54% of the global breeding population[3]. At Macquarie Island, the population declined through much of the last century, before growing slightly and once again declining more recently, with the overall population being negatively correlated with sea ice concentration[16]. In the Indian Ocean sector, populations on the Prince Edward archipelago, namely Marion Island, have declined by ~83% since the 1950s, with a more recent attrition rate of 5.8% annually[17,18]. At Îles Crozet, populations have decreased by 5.4% annually between 1970 and 1990[19–21]. At Îles Kerguelen, the population almost halved in size from 70,000 females in 1952 to 37,400 in 1987[20]. Following this, the population began to increase at almost 1% annually between 1987 and 2009[20,22]. However, recent evidence suggests that both Îles Crozet and Îles Kerguelen populations have entered a growth phase, increasing annually at 5.1% and 1.6%, respectively[23]. Available population data from Heard Island is over three decades old, but trends showed a decrease of ~50% from 1949 to 1985, which was linked to changes in sea-ice dynamics, followed by a period of relative stability from 1985 to 1992[24].

[1]British Antarctic Survey, High Cross, Madingley Road, Cambridge, CB3 0ET, UK. [2]Sea Mammal Research Unit, Scottish Oceans Institute, University of St Andrews, St Andrews, Fife, KY16 8LB, UK. [3]Centre for Ecology and Conservation, Faculty of Environment, Science and Economy, University of Exeter, Penryn Campus, Cornwall, TR10 9FE, UK. [4]School of Ocean Sciences, Prifysgol Bangor, Bangor University, Bangor, Gwynedd, LL57 2DG, UK. [5]These authors contributed equally: Connor. C. G. Bamford, Nathan Fenney, Jamie Coleman. ✉e-mail: conord48@bas.ac.uk

The expansion of highly pathogenic avian influenza viruses (HPAIV) across the globe, notably from clade 2.3.4.4b in 2020, has impacted mortality in wildlife populations globally[25]. Initially detected in Europe[26], this clade gained traction and crossed into North America[27,28], before spreading down into South America[29,30], culminating in mass mortalities of seabirds and marine mammals in 2022[29,31–34]. Background sampling of sites through the sub-Antarctic and Antarctic region showed that, as of March 2023, HPAIV had not been carried into this region[35]. However, in September 2023, the first suspected avian case was reported in brown skuas, *Stercorarius antarcticus*, on Bird Island, South Georgia[36], with confirmed mammalian occurrence of HPAIV several months later in both Antarctic fur seals *Arctocephalus gazella* and southern elephant seals[31,36]. As the season progressed, HPAIV was also confirmed in multiple other species island-wide[36]. In 2024, HPAIV was documented in the Indian Ocean, having been transferred from South Georgia to Îles Crozet and Îles Kerguelen[37].

Monitoring for HPAIV was conducted at South Georgia throughout the 2023/24 season, with opportunistic samples taken, where possible, at landing sites island-wide[36]. Transient observations, however, likely do not capture the true extent of the impact, as this virus affected southern elephant seals whilst ashore. Reports submitted from cruise ships[38] and from research activities (JC pers. comms.) suggest that the impact of this outbreak is likely comparable to the mass mortalities observed in the Argentinian Valdés population, which saw pup mortality of 97%[39]. Here, we present dedicated aerial survey data that indicates population decreases of comparable magnitudes at South Georgia, with surveys of the three largest breeding colonies in 2024 indicating an average reduction in female seal attendance of 47% (SD = 14.2%).

## Results

In 2024, seal breeding peaks were observed at St Andrews Bay on the 22nd October (4373 females ashore) and at Hound Bay on the 21st October (1154 females ashore), with subsequent counts showing declines. For the purpose of the HPAIV comparison, the number of females ashore on the comparison dates (Fig. 1) were 5.6% lower than the observed peak at St Andrews Bay (27th October), and 7.6% lower than the observed peak at Hound Bay (26th October). Peak dates were not available for Gold Harbour as, due to access issues, a time series could not be collected. Over both years, counts between the two observers were on average within 0.87% (SD = 1.1%) of each other.

In 2022, counts at St Andrews Bay, Gold Harbour and Hound Bay derived from ortho-rectified mosaics revealed that 6305 (95% CI: 6087.9–6528.1), 1599 (95% CI: 1538–1758) and 1901 (95% CI: 1784.2–2025.8) females were ashore at each of these respective beaches. Comparative counts from 2024 revealed an average reduction of 47% (SD = 14.2%) in the number of female seals present between 2022 and 2024, with only 4128 (95% CI: −3968.1 to 4323.9), 601 (95% CI: 538.2–673.8) and 1066 (95% CI: 982.6–1163.4) females present on each of the three colonies, respectively. When compared to long-term average counts (1958–2022), the observed number of females in 2024 constitutes an average decrease of 33.7% (SD = 2.3%) in the relative number of females present in 2024 (Table 1). If scaled to the entire island population at its last census[3], not accounting for population change over the past three decades, we predict about 53,000 females missed breeding in the 2024 season.

## Discussion

The last estimate of the South Georgia population, performed 30 years ago (1995), suggested that it represented over 50% of the global population of the species, with breeding sites located around the entirety of the island[3]. Here, we collected UAV aerial imagery at the three largest breeding beaches in both 2022 and 2024. Inter-observer variation was minimal, suggesting a high level of consistency, with variation stemming from minor adjustments in detection rather than systematic misclassification. Given this, we consider overhead counts to be a highly robust method for detecting population-level trends.

Arrival patterns of female southern elephant seals on breeding colony beaches are known to follow a distinct curve[3,6,7,10], which gradually increases to a peak with numbers tailing off afterwards. For a given date, counts revealed an average reduction of 47% between the absolute counts of female seals observed in 2024 compared to 2022. The three counted beaches represent 15.6% of the total population based on the 1995[3], which, if scaled in line with the last island-wide population estimate in 1995[3], a reduction of 47% equates to an estimated 53,000 female southern elephant seals failing to return for breeding following the confirmed arrival of HPAIV to South Georgia. Determining the return rate of these animals in future years remains crucial to understanding the dynamics and trends of this population. These findings suggest a substantial reduction in the number of adult females ashore in response to HPAIV.

At South Georgia, the severity of HPAIV's impact was not spatially uniform[36], similar to observations from the sub-Antarctic islands of the Indian Ocean[37], yet differing from the apparent uniform impact at Peninsula Valdés[39]. Smaller or more isolated breeding beaches experienced different apparent infection rates, with factors such as colony size, density, localised transmission rates and species composition influencing outcomes[36,37]. Therefore, assuming homogeneity may overestimate mortality at South Georgia. Further site-specific assessments are needed to refine population-wide estimates and capture potential regional variation in disease impact.

Regular long-term monitoring of this species has not been carried out at South Georgia; although stints of work have been conducted at Husvik during the 1980s/1990s[40–43] and work has been carried out intermittently in a potentially outlier population (due to its smaller size and location on an inner beach) at King Edward Cove over the past decade. The absence of regular monitoring on South Georgia's accessible breeding colony beaches complicates our understanding of the observed decline. However, where records exist, in the absence of HPAIV, interannual variation in the number of breeding females present at Gold Harbour indicates a deviation of ±3% between 1959 and 1964 (Table 1). Comparatively, where published, data from sites free of HPAIV and further afield also reflect interannual fluctuations within the same order of magnitude, for instance, between 1990 and 1997 at l'îles de la Possession, Îles Crozet, interannual variation averaged ±7.9%, and on the Courbet Peninsula, Îles Kerguelen, females present fluctuated within ±6.9% between adjacent years[21]. Similarly, between 1995 and 1997, the population at Peninsula Valdés reported interannual fluctuations of ±5.93%[44]. These comparative records support our assertion that the observed 47% decrease in females between 2022 and 2024 on South Georgia is atypical.

Due to accessibility, weather and slight differences in breeding phenology, it was not possible to fly all sites on the same day in consecutive years. In 2024, counts were taken either on the same date (St Andrews Bay) or one day later (Hound Bay and Gold Harbour) than in 2022 (Fig. 1).

**Fig. 1** | "Survey dates for flights of the three largest southern elephant seal colonies on South Georgia". "Dates of flights from the 2024 (black) and 2022 (red) seasons at the three largest southern elephant seal (*Mirounga leonina*) breeding colony beaches on South Georgia".

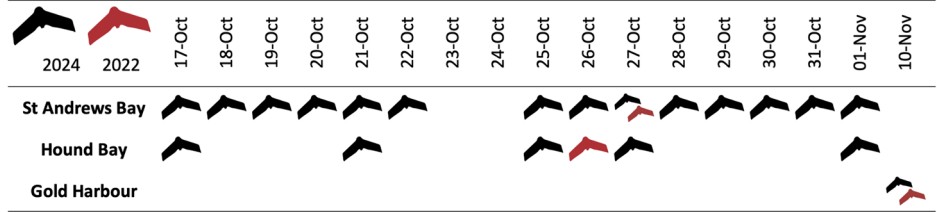

**Table 1 | Counts of adult female southern elephant seals (Mirounga leonina) present on breeding colony beaches from 1958 to 2024**

|  | St Andrews Bay | Gold Harbour | Hound Bay | Combined average |
|---|---|---|---|---|
| 1958 | – | 2611 | – | |
| 1959 | – | 2468 | 964 | |
| 1960 | – | 2243 | – | |
| 1961 | – | 2166 | 1270 | |
| 1963 | – | 1916 | – | |
| 1964 | – | 1833 | 2378 | |
| 1985 | 6198 | 4162 | | |
| 1995 | 5719 | 3332 | – | |
| 2019 | 6074 | – | 2122 | |
| 2022 | 6305 | 1599 | 1901 | |
| 2024 | 4128 | 601 | 1066 | |
| Long-term average | 6074 | 2481 | 1648 | |
| SD | 255 | 809 | 564 | |
| % decrease from long-term average | 32 | – | 35.3 | 33.7 (SD = 2.3) |
| % decrease since 2022 | 34.5 | 61.4 | 43.9 | 47 (SD = 14.2) |

"Numbers of adult female Southern elephant seals present on the beaches during the breeding season. Long-term averages were calculated from available counts from 1958 to 2022. Counts prior to 1995, obtained from ref. 57 and the BAS archives represent foot- and boat-based surveys and were corrected to the presumed peak of breeding (25th October). Post-1995 represent non-corrected counts, due to a lack of specific arrival curves, reported alongside their collection date. 1995[3] counts for St Andrews Bay and Gold Harbour were taken on the 17th and 18th October. In 2019[11], both sites were counted on the 25th of October. In 2022, counts were taken on the 26th October for Hound Bay, the 27th October for St Andrews Bay, and the 10th November for Gold Harbour. In 2024, counts were taken on the 27th October for both St Andrews Bay and Hound Bay, and on the 10th November for Gold Harbour. NB1: Counts for Gold Harbour in 2022 and 2024 were taken significantly after the peak of breeding and are therefore likely underestimates of the breeding count. NB2: Southern elephant seal harvesting at South Georgia was operational until the mid-1960s; counts in this era reflect a suppressed population baseline".

Whilst the later count in 2024 at Hound Bay will have slightly exacerbated the observed decrease, with the count being 7.6% lower than the observed peak (1154 females on the 21st October), this does not offset the magnitude of the observed reduction from 2022. These findings reinforce the atypical low signal in 2024, suggesting the one-day lag is unlikely to significantly affect the observed decline.

Several plausible hypotheses exist that could explain the observed decline between 2022 and 2024 at South Georgia. One explanation for the observed decline could be related to the strenuous conditions of the HPAIV-impacted 2023 breeding season, which resulted in numerous pup mortalities and/or abandonments alongside mortalities in both adult males and females (JC, Pers. Comm.). After losing their pup or abandoning them due to their own HPAIV-induced stress, females may have left the breeding colony beaches prematurely before oestrus, resulting in reduced copulation rates. Subsequently, this would lead to fewer pregnancies and, ultimately, fewer females returning to give birth in the following (monitored) season. Limited observations from Marion Island[45] suggest that at-sea copulation may occur, but its rarity implies that it is unlikely to offset reduced terrestrial breeding and low observed attendance in 2024. Alternatively, the emergence of HPAIV in the population may have triggered a shift in previous strong philopatry[46,47], with females returning to outlier colonies in 2024, dispersing the population more widely and lowering counts at the historically preferred beaches. If mating took place at sea, a rapid return to normal breeding behaviour and output might be expected. Further long-term investigation

into breeding dynamics is needed to understand phenology and recovery from this epidemic.

Finally, an unusual sea-ice anomaly in the South Atlantic during the 2023/2024 austral winter (National Snow and Ice Data Centre—https://nsidc.org/data/seaice_index) may have influenced elephant seal distribution and foraging, affecting their post-breeding/HPAIV recovery. However, given their wide-ranging movements[48] and known sea ice-association[49], the localised anomalous conditions in the South Atlantic are unlikely to have significantly impacted recovery or altered phenology.

While these hypotheses provide plausible explanations for some of the variation in population numbers, they alone, or in combination, are unlikely to fully account for the dramatic decline observed between 2022 and 2024. The temporal overlap of the arrival and prevalence of HPAIV in the elephant seal population during this period, coupled with the observed reductions, suggests a correlation that is too pronounced to be coincidental.

The long-term impact of the observed decline in the elephant seal population at South Georgia is yet to be determined. Research on adjacent populations has shown that female survival is a critical determinant of population growth[50,51]. Although we cannot be certain that all female absences are due to mortality, it is probable that a significant portion of these absent seals have perished. At Peninsula Valdés, HPAIV has had devastating effects, with abnormal mortality observed in 2023[39] and subsequent low attendance in 2024[52]. Demographic projection models for this population suggest that recovery could take decades[52], particularly because adult females, normally long-lived with low mortality, have a disproportionate effect on long-term population trends[51,53,54]. Under severe scenarios, full rebound is unlikely before the next century[52].

Due to the remoteness and relative inaccessibility of breeding colony beaches around much of South Georgia, accurately assessing mortality rates remains challenging. The average rate of female absence at South Georgia in 2024, whilst lower than the 67% observed at Peninsula Valdés, still remains significant. If the South Georgia population responds similarly to the modelled outlook at Peninsula Valdés, the future is bleak. This underscores the need for further research to understand the long-term impact on the South Georgia population, which, unlike Peninsula Valdés, lacks regular monitoring.

The average 47% decline observed between 2022 and 2024 across South Georgia's three largest breeding colony beaches is attributed to the direct impact of HPAIV. This dramatic drop contrasts sharply with historical interannual variations, which typically remain within 10%, both locally and in comparable Southern Ocean populations. While external environmental factors and behavioural shifts may have contributed, they alone cannot explain the severity of this decline.

The scale of this event may impact ongoing recruitment for the world's largest breeding population of southern elephant seals at South Georgia. To assess its short-term impact and determine whether the missing females represent true mortalities, follow-up monitoring, in line with recommendations[55], in 2025 and 2026, is imperative. To assess the long-term impacts on recruitment, sustained monitoring at the major breeding colony beaches is essential. Over recent years, the increased availability of high-resolution satellite imagery has provided an opportunity to examine recent trajectories. Integrating remotely sensed counts with ground data would allow an assessment of the consequences of HPAIV, enabling researchers to distinguish short-term fluctuations from enduring population-level impacts.

## Methods
### Aerial surveys
Aerial imagery from an uncrewed aerial vehicle (UAV) was collected in the 2022 and 2024 breeding seasons from the three largest breeding colony beaches on South Georgia. Both fieldwork seasons underwent review by the animal ethical approvals board at the British Antarctic Survey and were permitted under AWREB:1071 & 1109 in 2022 and 2024, respectively. These sampled years straddled the emergence of HPAIV on South Georgia. Flights in 2022 were part of an extended field campaign, which targeted

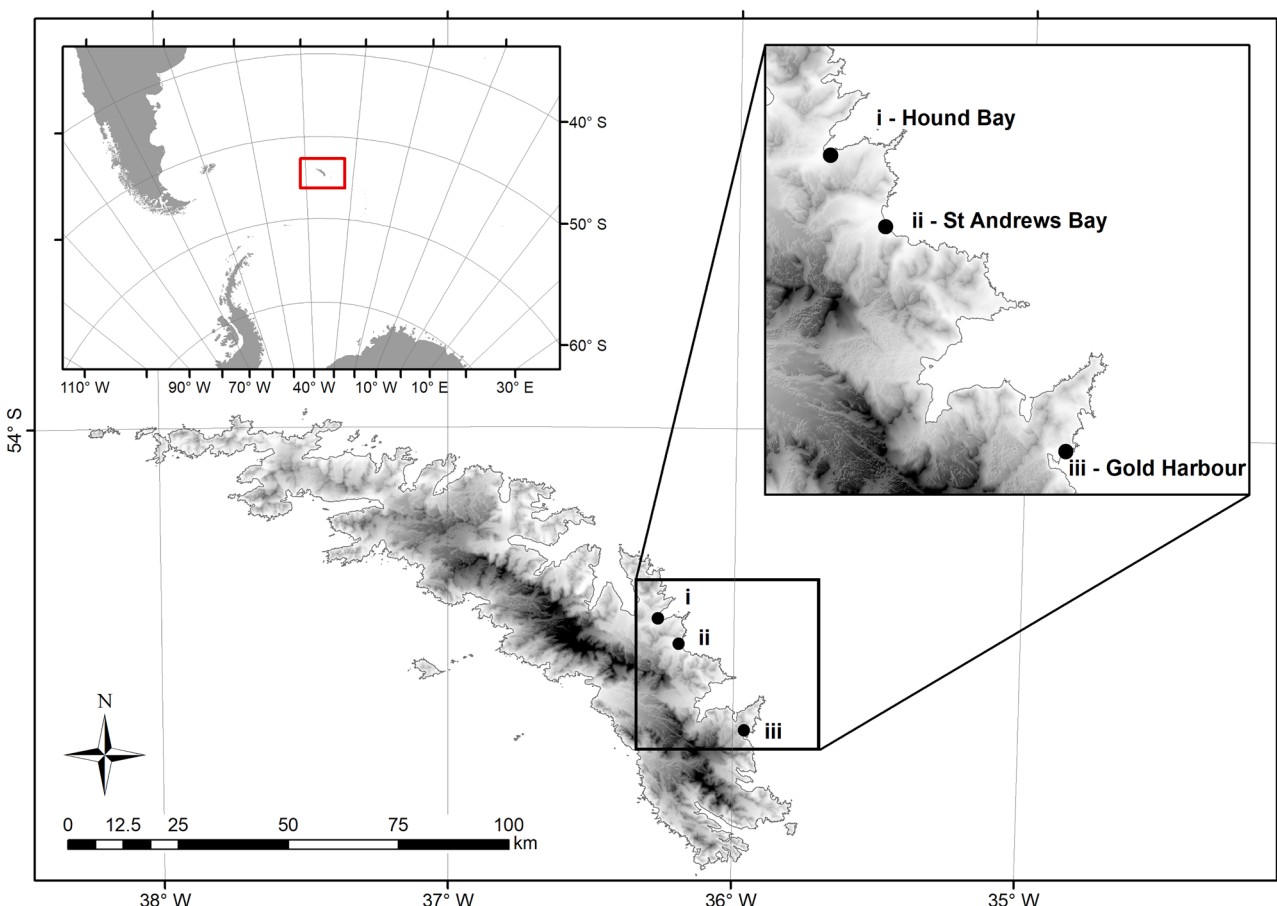

**Fig. 2 |** "Locations of the largest breeding colonies of southern elephant seal (Mirounga leonina) on South Georgia". "Sites of the three largest breeding colony beaches of Southern elephant seals (*Mirounga leonina*) on South Georgia (by total number of breeding females from the 1995 census[3]) where aerial imagery was collected in 2022 and 2024".

multiple locations and species around the South Georgia islands, meaning that sites were often only flown once. Conversely, in 2024, the field campaign specifically focused on southern elephant seals, which enabled sites to be flown multiple times successively (Figs. 1 and 2).

St Andrews Bay, Hound Bay and Gold Harbour (Fig. 1) were selected as they represent 15.6% of the island's southern elephant seal population at the last census[3]. St Andrews Bay, representing 7.5%, was flown on the 27th October in both 2022 and 2024; Gold Harbour, representing 4.4%, was flown on the 10th November in both 2022 and 2024; and Hound Bay, representing 3.7%, was flown on the 26th October 2022 and 27th October 2024 (Fig. 2).

In both seasons, flights were undertaken using a hand-launched, fixed-wing AgEagle eBee X UAV in fair weather with wind speeds <10 m/s. This UAV has a maximum flight time of ~90 min and was permitted for flights up to 182 m above surface level, although flights typically took ~15 min with a flight altitude of 90 m to achieve a suitable image resolution. The UAV carried a 24 megapixel (6000 × 4000 pixel) Aeria X RGB camera designed specifically for photogrammetry and mapping applications, along with a dual-band global navigation satellite system (GNNS) receiver used to determine the position of the image centres accurately to 1.5 cm.

**Image processing**

Simultaneously with each flight, a Trimble R9 GNSS base station collected precise point positioning (PPP) data following the methodology set out in ref. 56. Their data were used to maximise the quality of the exterior orientation of the downstream image processing using the online Canadian Spatial Reference System Precise Point Positioning (CSRS-PPP, v3) service, applying the International GNSS Service's (IGS) realisation of the International Terrestrial Reference Frame 2020 (ITRF2020). The precise location of the base station was then used to reprocess the eBee X's onboard GNSS data using a post-processed kinematics (PPK) workflow in eMotion (v 3.23). This yielded an updated latitude, longitude and height (above mean sea level, MSL) for each image.

Images were then processed following a Structure-from-Motion (SfM) photogrammetry workflow in Pix4D (v.4.9.0). Exterior orientations (x, y, z, $\Omega$, $\Phi$ and K—yaw, pitch and roll) of the camera during image capture were determined using both PPK solutions and common points within the images. These were used to derive a dense point cloud of the surface within the target area, with processing time being managed by downscaling images by a factor of 16. The dense point cloud was then orthorectified to produce a digital surface model (DSM) onto which the full-resolution original images were mapped; the effect of this was to remove distortion stemming from camera perspective and terrain shape, yielding a single orthorectified image mosaic for each flight.

In each of the ortho-rectified mosaics, adult females were counted, because they provide a reliable indicator of population trends in densely aggregated seal colonies[3,57] and are easily distinguishable from other demographics in overhead drone imagery[11]. Experienced observers, familiar with both the study species and the interpretation of overhead imagery, conducted the count. Using QGIS (v.3.22.16) and ESRI's ArcMap (v10.8), observers placed a point shapefile at the centre of each visible female along the three beaches. In 2022, each beach was counted twice by different observers, and, due to a high level of agreement between observers in 2022, in 2024 the counting process was streamlined, involving a first pass by one reviewer, followed by a second pass by an independent observer who reviewed the initial count, amending the shapefile for any false positives or negatives.

## Reporting summary

Further information on research design is available in the Nature Portfolio Reporting Summary linked to this article.

## Data availability

All UAV survey data are available on request from the NERC EDS UK Polar Data Centre. Data from the DPLUS109 2022/23 season for St Andrews Bay[58,59]; for Hound Bay[60]; and Gold Harbour[61]; these are under embargo until 31st December 2025. Data for the DPLUS214 2024/2025 season are available for St Andrews Bay[62]; Hound Bay[63]; and Gold Harbour[64]; these are under embargo until the 30th June 2026. Prior to these dates, please can interested parties contact the corresponding author to arrange access.

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

## Acknowledgements

This work was funded by the Biodiversity Challenge Fund Darwin Plus Main stage grants DPLUS109 and DPLUS214. Fieldwork was conducted under Government of South Georgia and the South Sandwich Islands Regulated Activity Permit Numbers 2022/021 and 2024/028 and ASSI permits P/314 and P/444 & 445 for the 2022 and 2024 seasons, respectively. These permits permitted BVLOS (Beyond Visual Line of Sight) UAV flights. Thanks also go to colleagues at BAS and partners at GSGSSI for their comments on this manuscript. We would also like to thank GSGSSI and the crew of MV Pharos SG for their logistical support during both field seasons. Thanks also go to Lindblad National Geographic Expeditions and the crew of the NG Explorer, who kindly supported the field team in 2022 and 2024. Both fieldwork seasons underwent review by the animal ethical approvals board at the British Antarctic Survey, with work approved and permitted under AWREB:1071 & 1109 in 2022 and 2024, respectively.

## Author contributions

This work was co-led by C.B., N.F. and J.C. Conceptualisation: C.B., N.F., J.C., L.H., M.F. and P.H. Methodology: C.B., N.F., J.C., P.F., M.F. and L.H. Data collection: 2024: C.B., N.F., J.C., C.F.C. and J.D. 2022: N.F. and J.C. Analysis: J.C. and N.F. in 2022, J.C., N.F. and C.B. in 2024. Drafting: C.B., with all authors contributing to the final review of this paper.

## Competing interests

The authors declare no competing interests.
