## [Transparent Peer Review file · Communications Biology]

Highly Pathogenic Avian Influenza Viruses (HPAIV) Associated with Major Southern Elephant Seal Decline at South Georgia

Corresponding Author: Dr Connor Bamford

Version 0:

Reviewer comments:

Reviewer #1

(Remarks to the Author)

Please see individual highlighted areas and comments in the PDF MS.

Two recent references that could be used in your paper may have been published after/during your submission

1. Clessin et al. bioRxiv 2025.02.25.640068; doi: <https://doi.org/10.1101/2025.02.25.640068>

2. Campagna, C., Condit, R., Ferrari, M., Campagna, J., Eder, E., Uhart, M., Vanstreels, R.E., Falabella, V. and Lewis, M.N., 2025. Predicting Population Consequences of an Epidemic of High Pathogenicity Avian Influenza on Southern Elephant Seals. *Marine Mammal Science*, p.e70009.

Reviewer #2

(Remarks to the Author)

This is a very useful manuscript, neat and tidy, with few typographical errors as far as I can see. It is very well written and properly analyzed. I conclude that (1) the manuscript does not have technical or conceptual flaws that should prohibit its publication, (2) the conclusions may not be original in a general sense as the catastrophic impact of the HPAIV has been described for the same species at Peninsula Valdés, Argentina (Campagna et al. 2024), but the conclusions are original for the South Georgia population, (3) no specific additional experiments would strengthen the case for publication, and the results presented are of immediate importance for both my discipline as well as for the public at large that keep/value animals. The outstanding feature of this manuscript is that it clearly points out the devastating effect of HPAIV on the elephant seals of South Georgia to the exclusion of other possible reasons for such a decline in population numbers. This fact is clearly articulated in the discussion and leaves no room for other interpretations of this catastrophe. I cannot comment on the Image Processing Method as it is outside of my field of expertise.

I have a few small suggestions and corrections.

1. Perhaps include Breed et al. (2023) which deals with the risk of the expansion of HPAI H5, and Dewar et al. (2023) that provides guidance to tourist operators and scientists in dealing with this pandemic.
2. Line 36 (and 103, 121, 175, 184, 194, 249, 275, 277, 290, 301): Beaches do not breed. Replace 'breeding beaches' with 'breeding colony beaches'
3. Line 50: Replace 'who' with 'that' or 'which'. We are dealing with animals, not persons.
4. Line 61: 'in the Southern Ocean'
5. Line 71: There is no such place as Crozet Island. It is an archipelago, so Crozet Islands would be a more appropriate usage. But see below.
6. Line 74: Strictly speaking one should use the terms 'Îles Crozet' and 'Îles Kerguelen' throughout.
7. Line 87: 'Antarctic fur seals *Arctocephalus gazella* and southern elephant seals'

8. Line 89: I have never heard of a 'snowy albatross' It must be a 'wandering albatross'
9. Line 92: I always reserve words such as 'however' for use in a discussion, not in the introduction. Start the sentence with 'transient observations'
10. Line 97: 'indicates at', why not just 'indicates'?
11. Line 121: Should be *Mirounga leonina*
12. Line 135: 'methodology in35, a Trimble' is awkward. Why not 'Simultaneous to each flight, a Trimble R9 GNSS base station collected precise point positioning (PPP) data35.'
13. Line 136: 'Their data' should be 'Their data' referring to paper 35.
14. Line 210: 'Possession Island, Îles Crozet' is correct. See in reference 17 the French name for this island.
15. Line 212: "females present fluctuated within +6.9% between adjacent years, and +9.85 and +7.89% biannually19". Not sure about the difference between 'adjacent years' and 'biannually'.
16. Line 234: Use 'Marion Island' throughout, not just 'Marion'.
17. Line 244: 'spatial removed' = 'spatially removed'?
18. Lines 334 – 492: The reference listing is a complete mess! There is no standard format. Titles appear in both capital letters and small case letters. Journal abbreviations are not standardized or not abbreviated at all. Species names are not in italics, and so on. See examples below:

Leguia M, et al. Highly pathogenic avian influenza A (H5N1) in marine mammals and seabirds in Peru. *Nature Communications* 14, 5489 (2023).

Ariyama N, et al. Highly Pathogenic Avian Influenza A(H5N1) Clade 2.3.4.4b Virus in Wild Birds, Chile. *Emerg Infect Dis* 29, 1842-1845 (2023).

Campagna C, Lewis M, Baldi R. BREEDING BIOLOGY OF SOUTHERN ELEPHANT SEALS IN PATAGONIA. *Mar Mamm Sci* 9, 34-47 (1993).

Laws R. The Elephant Seal (*Mirounga leonina*, Linn.): II. General, social and reproductive behaviour. *Scientific Reports Falkland Islands Dependencies Survey* 13:88, (1956).

References to the Report:

BREED, A., DEWAR, M., DODYK, L., KUIKEN, T., MATUS, R., SERAFINI, P. P., UHART, M., VANSTREELS, R. E. T., WILLE, M. (2023). Southward expansion of high pathogenicity avian influenza H5 in wildlife in South America: estimated impact on wildlife populations, and risk of incursion into Antarctica. OFFLU ad-hoc group on HPAI H5 in wildlife of South America and Antarctica. <https://www.offlu.org/wp-content/uploads/2023/08/OFFLU-statement-HPAI-wildlife-South-America-20230823.pdf>

CAMPAGNA, C., UHART, M., FALABELLA, V., CAMPAGNA, J., ZAVATTIERI, V., VANSTREELS, R.E.T. & LEWIS, M.N. (2024) Catastrophic mortality of southern elephant seals caused by H5N1 avian influenza. <https://doi.org/10.1111/mms.13101>

DEWAR M, WILLE M, GAMBLE A, et al. (2023) The risk of highly pathogenic avian influenza in the Southern Ocean: a practical guide for operators and scientists interacting with wildlife. *Antarctic Science* 35(6):407-414. doi:10.1017/S0954102023000342

MN Bester

Reviewer #3

(Remarks to the Author)

Using aerial drones, this paper examine the changes in southern elephant seals females during the breeding season in South Georgia between 2022 and 2024. The authors report a 46% decline in the number of breeding females on three of the main breeding beaches of South Georgia that they attribute to the Highly Pathogen Influenza Virus.

The subject and the collected data are highly important, but the paper, if it has to be published requires an in depth revision and I believe the authors could do a much better work.

First, the dynamic of the number of breeding females in South Georgia should be clearly presented. The peak date was neither mentioned or presented. I am certain this data is available for South Georgia, and the 2022 data in St Andrews Bay should allow determining that.

I missed a clear description of the cycle of occurrence of females the further away you are from the peak data the less female you have ashore. As the censuses, dates were quite variable between sites it is critical to have that information and to know how the numbers were corrected. However, I understood afterward that the seal number comparison were made for a given date for each colony. This has to be explained clearly as generally the numbers are referred to the total number of breeding taking into account the deviation of the census of breeding female from peak date and the maximum percentage of breeding female expected to be ashore on that date. Well-cited papers are available for South Georgia (Rothery & McCann 1987; Boyd et al. 1996).

On that point, exact number of individuals counted for each beach are provided, while the authors indicate that there is a 0.8% inter-observer variation in the number of female counted. I would have expect some kind of a confidence interval, even it is quite clear that the Drone censuses appear to be highly precise and I have no doubt on the validity of the census methodology implemented here.

However, one of the effect of the epizootic could be an early departure to sea of females which may have lost their pups and/or were sick. We know that in 2024 the HPAIV was still active in South Georgia. Therefore, even if the census are conducted at the same dates between two years, part of the decline reported could be to an early departure of some females. Some of those infected females may have died at sea, but others may have survived but would have left shore prior to mating increasing the proportion of non-pregnant females.

Detailed shore observation as well as satellite tracking data of females southern elephant seals revealed that non-pregnant females do not come back to shore during the breeding season but they will be back ashore to moult, remaining at sea for up to 10 to 11 months. This raises the question on how and where they breed.

So for these reasons it might be necessary to wait another year to be in a position to properly assess the long term impact of this HPAI event, but there is no doubt that the effect can be significant and this requires precise and long term monitoring. Although this was mentioned in the discussion, the global extrapolation to the decline detected on these three colonies to the whole South Georgia is subject to questions. Indeed according to my own experience on Kerguelen Island large local differences in infection rate and therefore pup mortality was observed between colonies.

In the discussion, the Campagna et al. 2024 paper is wrongly interpreted and this is extremely misleading. After checking, the HPAIV was, as I initially thought, only observed in 2023 in PV, they was no cases detected in 2024, but a spectacular decrease in the number of breeding females was seen in that year. So, I don't know how the authors could have understood that the HPAIV was active for three seasons and present pup mortality estimates for the 3 years revealing a lack of rigour.

For all these reasons, I cannot recommend the publication of this paper in its current form. However due to the importance of the data presented an in depth revision of this paper is necessary.

Version 1:

Reviewer comments:

Reviewer #1

(Remarks to the Author)

Please address to following

L1 change to Pathogenic as is commonly used in the abbreviation HPAIV, same at L27

L28 ..mortality in wildlife....

L48 delete "back"

L59 replace site with areas

L79-80 ...showed a decrease....., followed by a period of relative stability....

L81 ...clade spread into

L93 ...taken at landing sites....

At L95 you mention "moult" but no data are given for the moult in this paper and no reference provided for obs elsewhere

L106 delete which

L119 (Figures 1 and 2)

At L172 and 173 It is still not clear what the authors are saying here....the authors say numbers "peaked between the 22nd and 25th of Oct with 4373 seals ashore at St. Andrews Bay. But then go on to say this was -5.6% of the peak on the monitored dates. So is 4373 the peak number or not? Do you mean >90% of the seals had arrived on the 21st and >90% were still there on the 25th?

Please clarify what point you are making here.

L176was on average 0.87%.....

L187 space after (and before Table 1).

L189 decade, we predict about 53,000 females missed breeding in the 2024 season.

L198consider repeated aerial census to

L2032022. The three.....

L205if scaled to the

L206 ...island estimate in 1995, an estimated 53,000 female southern elephant seals failed to return for breeding following the confirmed.....

L226of the observed decline.

L229 space between (and Table ... as above

Reviewer #3

(Remarks to the Author)

I am happy with the revised version of this manuscript addresses the comments I have raised and I found it much clearer. The introduction and discussion were significantly improved. I have still some minor comments requested precision that I have included in the joined document.

Individual response to reviewer's comments in blue with new position line number included (where appropriate).

Reviewers' comments:

Reviewer #1 (Remarks to the Author):

Please see individual highlighted areas and comments in the PDF MS.

Two recent references that could be used in your paper may have been published after/during your submission

Added as suggested – ref 37: Clessin lines: 115, 321,324; Ref 54: Campagna 2025: 465,467,470.

1. Clessin et al. bioRxiv 2025.02.25.640068; doi: <https://doi.org/10.1101/2025.02.25.640068>

2. Campagna, C., Condit, R., Ferrari, M., Campagna, J., Eder, E., Uhart, M., Vanstreels, R.E., Falabella, V. and Lewis, M.N., 2025. Predicting Population Consequences of an Epidemic of High Pathogenicity Avian Influenza on Southern Elephant Seals. Marine Mammal Science, p.e70009.

...the confirmed arrival...

L113: initial text removed, but added later as suggested.

Please ensure the reader is clear on the pattern of female arrivals and departures.

The number of females arrivals and departures follows a Gaussian (normal) distribution. So numbers increase as rapidly as they decrease, Equally there are 50% ashore 2-3 weeks pre-peak.

L24: Amended for clarity.

to be consistent with later text please add "which includes the South Sandwich and Falkland Islands" to this sentence

L30: Amended as suggested.

? should read....and finally the Heard.....

L32: Corrected as suggested.

Hindell et al is not the primary source for this change to

van den Hoff, J., McMahon, C.R., Simpkins, G.R., Hindell, M.A., Alderman, R. and Burton, H.R., 2014. Bottom-up regulation of a pole-ward migratory predator population. Proceedings of the Royal Society B: Biological Sciences, 281(1782), p.20132842.

L59: Updated as suggested.

Add a short summary of what is known for the Heard island population which contributes about 30% to the Indian Ocean stock.

The latest ref is old but its the only one...

Slip, D.J. and Burton, H.R., 1999. Population status and seasonal haulout patterns of the southern elephant seal (*Mirounga leonina*) at Heard Island. Antarctic Science, 11(1), pp.38-47.

L66: Summary added to the relevant section in the introduction – thank you for your direction to the appropriate citation!

Delete this sentence.

This comparison adds nothing to the introduction.

Removed as suggested.

Shorten the sentence as some of the word are repetition from above
L119: Shortened and rephrased for clarity.

If I am not mistaken the pre1995 count data have been corrected to the 25th of October. Please correct the post-1995 count data in Table 1 to the same date using the proportion of females ashore in Fig. 1 of Boyd et al. 1996. Note: the long term averages may need to be recalculated in Table 1 for the numbers are corrected.

Throughout the MS: Thank you for this comment. We initially reached the same conclusion, however decided to retain the raw counts for the following three reasons:

1. Relevance of temporal standardisation for this paper's objectives: The primary aim of this study is to document the sharp interannual decline in female seal numbers between 2022 and 2024 following the emergence of HPAI in 2023. The counts compared in the analysis were taken on the same calendar date for two of the three beaches, and only one day apart for the third. Given this temporal consistency, the standardisation to a fixed peak date (e.g., the 27th October – as in Boyd et al. 1996) is not essential for interpreting the observed decline. Additionally, the scale of the observed decline between 2022 and 2024 far exceeds the variability associated with daily change in haul out counts, which we explicitly address in the revised manuscript with estimated peak declines percentages for the data available in 2024.
2. Limitation in applying Boyd et al. 1996 corrections to the presented data: While Boyd et al. present an average haul-out peak of around the 25th-27th (Fig1, Table 1), their correction model is built upon CDFs specific to South Georgia in 1995, with site-specific parameters derived from repeat in-season counts from different beaches to the ones presented in the current manuscript. Importantly here, the raw data or fitting parameters needed to re-run their model to apply onto our data are not available. All we have access to is uncorrected, individual beach counts from their survey (as reported in our Table 1). As Boyd et al. 1996 note, "The form taken by the distribution of numbers hauled-out depends on the values of s and S " (p239), and even small shifts in these parameters (i.e., differing peak dates, or haul out duration) can cause significant variation in estimated totals (see. Their Fig 5). Without site-specific time series for each haul out year and beach, applying a generalised correction for 1995 to our recent data would introduce greater uncertainty, not less. We have removed the Gold Harbour counts in 2024 along with their contribution to the long-term average, and only make comparisons to the two other beaches, whose counts in 2024 alongside the records pre-2022, were taken or corrected close to the peak of breeding.
3. Applying meaningful corrections based on haul-out timing would not meaningfully change the observed patterns. If anything, applying the Boyd et al. 1996 adjustment would increase the presented count in Table 1 (since the counts in 1995 were made pre-peak), thus amplifying the observed decline when compared to 2024. Counts in 2019, presented in Table 1, were already made on the 25th October, and thus fall within the Boyd et al. correction window. For 2022, counts at St Andrews Bay and Gold harbour were taken on the same calendar date post peak, suggesting any post-peak declines in should be broadly comparable. For Hound Bay, where the 2022 count is one day earlier than in 2024, even using our originally presented daily decline estimates of between -3.3% (SD 1.9%) to -5.5% (SD 9.9%), the timing offset cannot account for the magnitude of the observed drop in female numbers.

In sum, any error introduced by attempting to standardise using assumptions from a different year and location would likely exceed the theoretical gains, which are already evident through logical interpretation of the data we provide. For these reasons, we have chosen to retain the raw counts as collected, while explicitly acknowledging and quantify post-peak declines where available (i.e., for St Andrews and Hound Bay in 2024). We believe that this approach provides a more transparent and methodologically sound basis for the interannual comparisons central to our conclusions.

Repetition from L53-55 but with different references,
Is a difference being implied here? If so what is it
Section removed to improve flow.

The count data that were collected over 14 days would have captured the increase, the peak and the decrease in numbers of females.

What does daily reduction rate actually mean here? Were there 3.3% fewer females arriving/departing SG per day in 2024 compared with earlier census years or something else?

Is daily reduction rate the same as post-peak attrition rate? They seem different

L226: Section rephrased for clarity with specific & decreases from the peak provided to the reader. References to daily attrition rates removed.

I think you mean "the three largest breeding beaches"

L242: Corrected as suggested.

the "confirmed" arrival of HPAI at SG in 2023

L275: Corrected as suggested.

Im not sure decline is the right word to use here, suggest a change to

... a substantial reduction in numbers of adult females ashore in response to HPAIV,

L278: Corrected as suggested.

According to Campagna et al. (2023) the effects were "virtually identical" across the three beaches they examined at PV. Some sites were 50 km apart.

L280: Section has been rephrased to improve the clarity for the reader, and supporting evidence provided. At South Georgia, HPAIV's impact was not uniform making an interesting comparison to PV. At SG, the larger beaches / colonies were impacted more severely than smaller ones. It's hypothesised that colony size, density and species composition might be playing a role here.

Bennison, A. et al. A case study of highly pathogenic avian influenza (HPAI) H5N1 at Bird Island, South Georgia: the first documented outbreak in the subantarctic region. *Bird Study*, 1-12, doi:10.1080/00063657.2024.2396563 (2024).

Im not sure any area is isolated from HPAI

It has crossed oceans from PV to Kerguelen

L282: Comment clarified with additional supporting information added.

...in the absence of HPAIV....

L295: Amended and included for clarity.

...sites free of HPAIV and further afield....

~L297: Section amended.

The PV SES colony also experienced high pup mortality in the same HPAI year (2023) and low adult female return rates in the year following HPAI (2024) see Campagna et al. 2025.

67% fewer females returned to PV, a figure comparable to your almost 50%
I think it is pretty clear a 50% and 67% reduction in adult females ashore at two breeding locations in the year following a HPAI year is clearly attributable to HPAI.
Im not sure all of the discussion below about potential contributor to the decrease is actually needed. Overall at SG variability is in the order of 3% pa. And as the authors note that about what it is measured at most other locations in the absence of HPAI.
Campagna, C., Condit, R., Ferrari, M., Campagna, J., Eder, E., Uhart, M., Vanstreels, R.E., Falabella, V. and Lewis, M.N., 2025. Predicting Population Consequences of an Epidemic of High Pathogenicity Avian Influenza on Southern Elephant Seals. Marine Mammal Science, p.e70009.
L439: 2025 reference which was published post-submission has now been integrated.

The proof seems fairly conclusive that HPAI was the principal factor in the reduction in numbers of females beyond pre-defined natural variability.
Im not convinced the discussion highlighted in green below is actually required to assure the reader that HPAI was the main contributor to the observed decrease in numbers of females ashore. After consideration each factor is dismissed anyway.
Discussion section: Efforts have been made to significantly reduce the length of the highlighted sections of the discussion, whilst also maintaining the discussive elements commented on my other reviewers. We hope that these changes will satisfy both reviewers and improve the narrative for the reader.

Not clear what period this "observed average decline" refers to
Is it for the period between 1956 and 2022 or 1956 to 2024 or something else
L300: Clarified – 2022 to 2024 observed decline.

...explanation for the observed....
L366 onwards: Amended as suggested – check as this section might be toned down / removed.

Could you make a comment on whether dead adult males and females were seen on the beach at SG or just pups?
At the moment the previous sentence suggests it was pups only.
L369: Comment added as requested.

I think reference to a recent paper from Campagna et al that models the consequences of HPAI on elephant seals would reduce the length of the discussion and speculation.
Campagna et al
<https://doi.org/10.1111/mms.70009>
Ref 54 added throughout: Section amended, and new demographic models incorporated into the discussion.

refer to Campagna et al for a model
<https://onlinelibrary.wiley.com/doi/10.1111/mms.70009>
Section amended and new demographic models incorporated into the discussion.

...is attributed to the direct....
L460: Corrected as suggested.

Table1: For comparisons, could all these numbers be corrected to the 25th of October
Table 1: Please refer to above response to this comment.

Reviewer #2 (Remarks to the Author):

This is a very useful manuscript, neat and tidy, with few typographical errors as far as I can see. It is very well written and properly analyzed. I conclude that (1) the manuscript does not have technical or conceptual flaws that should prohibit its publication, (2) the conclusions may not be original in a general sense as the catastrophic impact of the HPAIV has been described for the same species at Península Valdés, Argentina (Campagna et al. 2024), but the conclusions are original for the South Georgia population, (3) no specific additional experiments would strengthen the case for publication, and the results presented are of immediate importance for both my discipline as well as for the public at large that keep/value animals. The outstanding feature of this manuscript is that it clearly points out the devastating effect of HPAIV on the elephant seals of South Georgia to the exclusion of other possible reasons for such a decline in population numbers. This fact is clearly articulated in the discussion and leaves no room for other interpretations of this catastrophe. I cannot comment on the Image Processing Method as it is outside of my field of expertise.

I have a few small suggestions and corrections.

1. Perhaps include Breed et al. (2023) which deals with the risk of the expansion of HPAI H5, and Dewar et al. (2023) that provides guidance to tourist operators and scientists in dealing with this pandemic.

Ref 30 & 57: Thanks for these suggestions – citations now incorporated into relevant sections in the introduction and discussion.

2. Line 36 (and 103, 121, 175, 184, 194, 249, 275, 277, 290, 301): Beaches do not breed. Replace 'breeding beaches' with 'breeding colony beaches'

Corrected in all instances.

3. Line 50: Replace 'who' with 'that' or 'which'. We are dealing with animals, not persons.

Corrected.

4. Line 61: 'in the Southern Ocean'

Corrected.

5. Line 71: There is no such place as Crozet Island. It is an archipelago, so Crozet Islands would be a more appropriate usage. But see below.

Corrected and switched to Îles throughout.

6. Line 74: Strictly speaking one should use the terms 'Îles Crozet' and 'Îles Kerguelen' throughout.

Corrected and switched to Îles for each throughout.

7. Line 87: 'Antarctic fur seals *Arctocephalus gazella* and southern elephant seals'

L78: Corrected.

8. Line 89: I have never heard of a 'snowy albatross' It must be a 'wandering albatross'

Correct – 'snowy' is the new nomenclature for 'wandering' albatross and is consistent with the cited literature. I believe that this naming shift was part of an effort to distinguish the physically larger South Georgia population from other wandering populations. However, this section has been removed from the resubmitted MS

9. Line 92: I always reserve words such as 'however' for use in a discussion, not in the introduction. Start the sentence with 'transient observations'

L97: Thank you for the observation. We agree that position and tone are important in the introduction. In response, we have revised the sentence to begin with "transient observations" as suggested, while retaining "however" in the middle of the sentence to preserve the intended contrast. We hope that this placement softens the transitional tone and ensures clarity for the reader, without being overly discussive in the introduction.

10. Line 97: 'indicates at', why not just 'indicates'?

L103: Amended as suggested.

11. Line 121: Should be *Mirounga leonina*

Fig 2: Thank you for spotting this typo! Corrected.

12. Line 135: 'methodology in35, a Trimble' is awkward. Why not 'Simultaneous to each flight, a Trimble R9 GNSS base station collected precise point positioning (PPP) data35.
L160: Sentence amended to increase clarity.

13. Line 136: 'There data' should be 'Their data' referring to paper 35.

L161: Corrected.

14. Line 210: 'Possession Island, Îles Crozet' is correct. See in reference 17 the French name for this island.

L299: Corrected as suggested

15. Line 212: "females present fluctuated within +6.9% between adjacent years, and +9.85 and +7.89% biannually19". Not sure about the difference between 'adjacent years' and 'biannually'.

L300: Simplified for clarity – only adjacent years (i.e., 1990 and 1991 or 1992 and 1993) provided in the revision.

16. Line 234: Use 'Marion Island' throughout, not just 'Marion'.

Corrected throughout.

17. Line 244: 'spatial removed' = 'spatially removed'?

Removed from MS, but thank you for spotting this.

18. Lines 334 – 492: The reference listing is a complete mess! There is no standard format. Titles appear in both capital letters and small case letters. Journal abbreviations are not standardized or not abbreviated at all. Species names are not in italics, and so on. See examples below:

Corrected and standardised.

Leguia M, et al. Highly pathogenic avian influenza A (H5N1) in marine mammals and seabirds in Peru. *Nature Communications* 14, 5489 (2023).

Ariyama N, et al. Highly Pathogenic Avian Influenza A(H5N1) Clade 2.3.4.4b Virus in Wild Birds, Chile. *Emerg Infect Dis* 29, 1842-1845 (2023).

Campagna C, Lewis M, Baldi R. BREEDING BIOLOGY OF SOUTHERN ELEPHANT SEALS IN PATAGONIA. *Mar Mamm Sci* 9, 34-47 (1993).

Laws R. The Elephant Seal (*Mirounga leonina*, Linn.): II. General, social and reproductive behaviour. *Scientific Reports Falkland Islands Dependencies Survey* 13:88, (1956).

References to the Report:

Added to reference list – thank you for these suggestions.

BREED, A., DEWAR, M., DODYK, L., KUIKEN, T., MATUS, R., SERAFINI, P. P., UHART, M., VANSTREELS, R. E. T., WILLE, M. (2023). Southward expansion of high pathogenicity avian influenza H5 in wildlife in South America: estimated impact on wildlife populations, and risk of incursion into Antarctica. OFFLU ad-hoc group on HPAI H5 in wildlife of South America and Antarctica. <https://www.offlu.org/wp-content/uploads/2023/08/OFFLU-statement-HPAI-wildlife-South-America-20230823.pdf>

CAMPAGNA, C., UHART, M., FALABELLA, V., CAMPAGNA, J., ZAVATTIERI, V., VANSTREELS, R.E.T. & LEWIS, M.N. (2024) Catastrophic mortality of southern elephant seals caused by H5N1 avian influenza. <https://doi.org/10.1111/mms.13101>

DEWAR M, WILLE M, GAMBLE A, et al. (2023) The risk of highly pathogenic avian influenza in the Southern Ocean: a practical guide for operators and scientists interacting with wildlife. Antarctic Science 35(6):407-414. doi:10.1017/S0954102023000342

MN Bester

Reviewer #3 (Remarks to the Author):

Using aerial drones, this paper examine the changes in southern elephant seals females during the breeding season in South Georgia between 2022 and 2024. The authors report a 46% decline in the number of breeding females on three of the main breeding beaches of South Georgia that they attribute to the Highly Pathogen Influenza Virus.

The subject and the collected data are highly important, but the paper, if it has to be published requires an in depth revision and I believe the authors could do a much better work.

First, the dynamic of the number of breeding females in South Georgia should be clearly presented. The peak date was neither mentioned or presented. I am certain this data is available for South Georgia, and the 2022 data in St Andrews Bay should allow determining that.

I missed a clear description of the cycle of occurrence of females the further away you are from the peak data the less female you have ashore. As the censuses, dates were quite variable between sites it is critical to have that information and to know how the numbers were corrected. However, I understood afterward that the seal number comparison were made for a given date for each colony. This has to be explained clearly as generally the numbers are referred to the total number of breeding taking into account the deviation of the census of breeding female from peak date and the maximum percentage of breeding female expected to be ashore on that date. Well-cited papers are available for South Georgia (Rothery & McCann 1987; Boyd et al. 1996).

On that point, exact number of individuals counted for each beach are provided, while the authors indicate that there is a 0.8% inter-observer variation in the number of female counted. I would have expect some kind of a confidence interval, even it is quite clear that the Drone censuses appear to be highly precise and I have no doubt on the validity of the census methodology implemented here.

L258: Section clarified and CI's added to the reported counts.

However, one of the effect of the epizootic could be an early departure to sea of females which may have lost their pups and/or were sick. We know that in 2024 the HPAIV was still active in South Georgia. Therefore, even if the census are conducted at the same dates between two years, part of the decline reported could be to an early departure of some females. Some of those infected females may have died at sea, but others may have survived but would have left shore prior to mating increasing the proportion of non-pregnant females.

Detailed shore observation as well as satellite tracking data of females southern elephant seals revealed that non-pregnant females do not come back to shore during the breeding season but they will be back ashore to moult, remaining at sea for up to 10 to 11 months.

This raises the question on how and where they breed. Agreed, this warrants further investigation at South Georgia. We bring in the element of at-sea copulation (L374) in the discussion but note that this cannot fill the shortfall in breeding in 2024. Definitely raises the question around the biological pressure on females to return to breed, and when that will kick in.

So for these reasons it might be necessary to wait another year to be in a position to properly assess the long term impact of this HPAI event, but there is no doubt that the effect can be significant and this requires precise and long term monitoring.

We agree with the reviewer wholeheartedly here. However, long-term, large scale monitoring of elephant seals at South Georgia is not routinely conducted by BAS, nor any other institute. We hope that the presentation of these observations and results will aid in the establishment of such a programme, which in the future can hopefully delve into these questions.

Although this was mentioned in the discussion, the global extrapolation to the decline detected on these three colonies to the whole South Georgia is subject to questions. Indeed according to my own experience on Kerguelen Island large local differences in infection rate and therefore pup mortality was observed between colonies.

It's interesting that both the sub-Antarctic islands have observed differing infection rates/mortality/severity of the outbreak island-wide, yet the impact was uniform in PV (Reviewer 2). In an effort to address the concerns with the island-wide extrapolation in line with the 1995 census, we've toned down the emphasis on this figure and only reference it in the discussion not abstract.

In the discussion, the Campagna et al. 2024 paper is wrongly interpreted and this is extremely misleading. After checking, the HPAIV was, as I initially thought, only observed in 2023 in PV, they was no cases detected in 2024, but a spectacular decrease in the number of breeding females was seen in that year. So, I don't know how the authors could have understood that the HPAIV was active for three seasons and present pup mortality estimates for the 3 years revealing a lack of rigour.

L439: We thank the reviewer for bringing this misinterpretation to our attention and have corrected this error in these revisions.

For all these reasons, I cannot recommend the publication of this paper in its current form. However due to the importance of the data presented an in depth revision of this paper is necessary.

Dramatic decline in the world's largest population of
southern elephant seals attributed to the arrival of High
Pathogenicity Avian Influenza Viruses (HPAIV) on
South Georgia

Bamford, C.C.G.^{1*},
Fenney, N.¹,
Coleman, J.¹,
Fox-Clarke, C.¹,
Dickens, J.D.¹,
Fedak, M.³
Fretwell, P.¹
Hückstädt, L.⁴
Hollyman, P.^{2, 1}

1. British Antarctic Survey, High Cross, Madingley Road, Cambridge, CB3 0ET, United
Kingdom
2. School of Ocean Sciences, Prifysgol Bangor, Bangor University, Bangor, Gwynedd,
LL57 2DG
3. Sea Mammal Research Unit, Scottish Oceans Institute, University of St Andrews, St
Andrews, Fife, UK, KY16 8LB
4. Centre for Ecology and Conservation, Faculty of Environment, Science and Economy,
University of Exeter, Penryn Campus, Cornwall, TR10 9FE.

* Corresponding author

Abstract

The emergence of high pathogenicity avian influenza viruses (HPAIV) has dramatically
impacted avian and marine mammals worldwide. In South America, mass mortalities of
southern elephant seals *Mirounga leonina* were observed from 2022 onwards. HPAIV spread
into the sub-Antarctic in 2023, where it has impacted multiple species and populations.
However, the remoteness of these islands has hindered our understanding of the true scale of
this virus' impact. Here we present compelling evidence of HPAIV's dramatic effect on the
number of breeding females at the world's largest population of southern elephant seals at
South Georgia. Following the arrival of HPAIV in 2023, we observed a decrease of 46.6%
(SD = 13.5%) in the number of females present at the three largest breeding beaches in 2024.
Extrapolated to the entire population using the most recent estimates of the island's
population from the 1995 census, this reduction corresponds to ~62,000 absent females
following the virus's introduction. The removal of such a substantial proportion of females
will likely have a profound impact on the recruitment and status of this population. This
situation underscores the need for ongoing and intensified monitoring of this species in the
forthcoming seasons and into the future.

Introduction

Southern elephant seals *Mirounga leonina* are the largest of the pinnipeds, and are a major
predator with a circumpolar distribution in the Southern Hemisphere¹. Southern elephant
seals breed annually and come ashore on sub-Antarctic islands² at the end of the Austral
winter where they form dense colonies comprised of competitive harems on beaches^{3, 4, 5}.
Large males haul out first, towards the end of September and early October, followed by
pregnant females, who give birth ~3-5 days post-arrival^{6, 7}. Pups are then weaned
approximately 22-23 days post-partum, with females coming back into oestrus several days
prior to weaning^{6, 8, 9}. The number of females decrease rapidly after the peak as females
depart post weaning and copulation, leaving approximately 50% of the maximum ashore two-
to-three weeks post-peak^{3, 10}.

Four genetically distinct populations have been identified within the Southern Hemisphere:
the Peninsula Valdés population in Argentina; (ii) the South Georgia population in the South
Atlantic; the Macquarie population in the South Pacific; and finally Heard and Kerguelen
populations, which includes the Crozet and Prince Edward archipelagos, in the south Indian

Ocean^{11, 12}. These four broad sites comprise the principal breeding locations for this species
around the Southern Ocean.

Population trajectories are varied throughout the Southern Ocean. The Peninsula Valdés
population in Argentina has been growing at between 1 to 3.4% annually for the last five
decades¹³. At its last census in 1995, the South Georgia population was deemed to be stable
and accounted for ~54% of the global breeding population³. At Macquarie Island, the
population declined through much of the last century, before growing slightly and once again
declining more recently, with the overall population being negatively correlated with sea ice
concentration¹⁴. In the Indian Ocean sector, populations on the Prince Edward archipelago,
namely Marion Island, have declined by ~83% since the 1950s, with a more recent attrition
rate of 5.8% annually^{15, 16}. On Crozet Island, populations have decreased by 5.4% annually
between 1970 and 1990^{17, 18, 19}. At Kerguelen islands, the population almost halved in size
from 70,000 females in 1952 to 37,400 in 1987¹⁸. Following this, the population began to
increase at almost 1% annually between 1987 and 2009^{18, 20}. However, recent evidence
suggests that both Crozet and Kerguelen populations have entered a growth phase, increasing
annually at 5.1% and 1.6%, respectively²¹. With this growth, the combined populations in the
Indian Ocean sector are close to rivalling the last counts for South Georgia.

The expansion of high pathogenicity avian influenza viruses (HPAIV) across the globe,
notably from clade 2.3.4.4b in 2020, has impacted wildlife populations globally²². Initially
detected in Europe²³, this clade gained traction and crossed into North America^{24, 25}, before
spreading down into South America²⁶, culminating in mass mortalities of seabirds and marine
mammals in 2022^{26, 27, 28, 29, 30}. Background sampling of sites through the sub-Antarctic and
Antarctic region showed that, as of March 2023, HPAIV had not been carried into this region
85³¹. However, in September 2023, the first suspected avian case was reported in brown skuas,
*Stercorarius antarcticus*, on Bird Island, South Georgia³², with confirmed mammalian
occurrence of HPAIV several months later in both Antarctic fur *Arctocephalus gazella* and
southern elephant seals^{27, 32}. As the season progressed, HPAIV was also confirmed in gentoo
penguins *Pygoscelis papua*, and snowy albatross *Diomedea exulans* at South Georgia³².

Monitoring for HPAIV was conducted at South Georgia throughout the 2023/24 season³²,
with opportunistic samples taken at landing sites island-wide. However, as this virus
impacted southern elephant seals over the whole season, transient observations likely do not

capture the true extent of the impact. Reports submitted from cruise ships and from research
 activities³³, suggest that the impact of this outbreak is likely comparable to the mass
 mortalities observed in the Argentinian Valdés population, which saw pup mortality upwards
 of 70%³⁴. Here, we present dedicated aerial survey data that indicates at population decreases
 of comparable magnitudes at South Georgia.

Methods

Aerial surveys

Aerial imagery from an unpowered aerial vehicle (UAV) was collected over two seasons from
 the three largest breeding beaches in 2022 and 2024, straddling the emergence of HPAIV on
 South Georgia. Flights in 2022 were part of an extended field campaign, which targeted
 multiple locations and species around the South Georgia islands, meaning that sites were
 often only flown once. Conversely, in 2024, the field campaign specifically focused on
 southern elephant seals, which enabled sites to be flown multiple times successively (Figure
 1).

*Figure 1 – Dates of flights from the 2024 (black) and 2022 (red) seasons at the three largest southern elephant*
 *seal breeding beaches on South Georgia.*

**Three beaches (St Andrews Bay, Hound Bay and Gold Harbour, Figure 1) were selected as**
 **they represent the three largest rookeries of southern elephant seals on South Georgia,**
 **accounting for 15.6% of the island’s population at last census³.** St Andrews Bay, representing
 7.5%, was flown on the 27th October in both 2022 and 2024; Gold Harbour, representing
 4.4%, was flown was on the 10th November in both 2022 and 2024; and Hound Bay,
 representing 3.7%, was flown on the 26th October 2022 and 27th October 2024 (Figure 2).

 *Figure 2 – Sites of the three largest breeding beaches of Southern elephant seals (Mirounga leonine) on South*
 *Georgia islands (by total number of breeding females from the 1995 census³) where aerial imagery was collected*
 *in 2022 and 2024.*

 In both seasons, flights were undertaken using a hand-launched, fixed-wing AgEagle eBee X
 UAV in fair weather with wind speeds <10 m/s. This UAV has a maximum flight time of ~90
 minutes and was permitted for flights up to 182m above surface level, although flights
 typically took ~15 minutes with a flight altitude of 90m to achieve a suitable image
 resolution. The UAV carried a 24 mega-pixel (6000 × 4000 pixel) Aeria X RGB camera
 designed specifically for photogrammetry and mapping applications along with a dual-band
 global navigation satellite system (GNSS) receiver used to determine the position of the
 image centres accurate to 1.5cm.

Image processing

Simultaneous to each flight, and following the methodology in³⁵, a Trimble R9 GNSS base
 station collected precise point positioning (PPP) data. These data were used to maximise the
 quality of the exterior orientation of the downstream image processing using the online
 Canadian Spatial Reference System Precise Point Positioning (CSRS-PPP, v3) service

applying the International GNSS Service's (IGS) realisation of the International Terrestrial
Reference Frame 2020 (ITRF2020). The precise location of the base station was then used to
reprocess the eBee X's onboard GNSS data using a post-processed kinematics (PPK)
workflow in eMotion (v 3.23). This yielded an updated latitude, longitude and height (above
mean sea level, MSL) for each image.

Images were then processed following a Structure-from-Motion (SfM) photogrammetry
workflow in Pix4D (v.4.9.0). Exterior orientations (x , y , z , Ω , Φ and K – yaw, pitch and roll)
of the camera during image capture were determined using both PPK solutions and common
points within the images. These were used to derive a dense point cloud of the surface within
the target area, with processing time being managed by downscaling images by a factor of 16.
The dense point cloud was then orthorectified to produce a digital elevation model (DEM)
onto which the full-resolution original images were mapped; the effect of this was to remove
distortion stemming from camera perspective and terrain shape yielding a single ortho-
rectified image mosaic for each flight.

In each of the ortho-rectified mosaics, adult females were counted in QGIS (v.3.22.16) by
experienced observers, familiar with both interpreting overhead imagery and the study
species. Adult females were counted as they provide a viable means of assessing trends in
densely aggregated seal colonies^{3, 36} and are easily distinguishable from other demographics
in overhead drone imagery³⁷. A point shapefile was positioned on the centre of each visible
animal, with a first-pass count conducted by one observer and reviewed by a second to check
for erroneous omissions from the count; variation between observer counts was <0.8%.

Results

In 2022, counts at St Andrews Bay, Gold Harbour and Hound Bay derived from ortho-
rectified mosaics revealed that 6,305, 1,550 and 1,901 females were ashore at each of these
respective beaches. Comparative counts from 2024 revealed an average reduction of 46.6%
(SD = 13.5%) in the number of female seals present between 2022 and 2024, with only
4,128, 601 and 1,066 females present on each of the three colonies. When compared to long-
term average counts (1958 to 2022), the observed number of females in 2024 constitutes an
average decrease of 47.7% (SD = 24.3%) in the relative number of females present in 2024 (
Table 1). If scaled to the entire island population at its last census³, not accounting for

population change over the past three decades, this places this reduction in the order of
62,000 missing females.

Arrival patterns of female southern elephant seals on breeding beaches are known to follow a
distinct curve^{6,7}, which gradually increases to a peak with numbers tailing off afterward. The
availability of counts from 14 of 16 consecutive days at St Andrews Bay in 2024 enabled a
daily reduction rate of -3.3% (SD = 1.9%) females to be estimated. For Hound Bay, where
consecutive counts were less available, the average daily reduction rate in the number of
female seals was -5.5% (SD = 9.9%).

Discussion

The last estimate of the South Georgia population, 30 years ago (1995), suggested that it
represented over 50% of the global population of the species, with breeding sites located
around the entirety of the island³. Here we collected UAV aerial imagery at the three largest
sites in both 2022 and 2024. Counts revealed an average reduction of 46.6% between
absolute counts of female seals observed in 2024 compared to 2022; or a 47.7% reduction if
counts from 2024 are compared to averaged counts from 1958 – 2022. Scaled to the whole
island according to the last available population estimate from the 1995 census³, a reduction
of 46.6% equates to an estimated 62,000 missing females following the arrival of HPAIV to
South Georgia.

While these findings suggest a substantial decline, they are based on 15.6% of the total
breeding population³, albeit from the three largest breeding sites. The impact of HPAI is
unlikely to be spatially uniform, and assuming homogeneity may overestimate mortality.
Smaller or more isolated colonies may have experienced different infection rates and
outcomes. Further site-specific assessments are needed to refine population-wide estimates
and capture potential regional variation in disease impact.

Regular long-term monitoring of this species has not been carried out at South Georgia;
although stints of work have been conducted at Husvik during the 1980/90s^{38,39,40,41} and
work has been carried out intermittently in a potentially outlier population (due to its smaller
size and location on an inner beach) at King Edward Cove over the past decade (unpublished
BAS long term monitoring data). The absence of regular monitoring on South Georgia's

accessible breeding beaches complicates our understanding of the atypicality of the observed
decline. However, where records exist, interannual variation in the number of breeding
females present at Gold Harbour indicates a deviation of $\pm 3\%$ between 1959 and 1964 (Table
1). Comparatively, where published, data from sites further afield also reflect interannual
fluctuations within the same order of magnitude, for instance between 1990 and 1997 at
Possession Island, Îles Crozet interannual variation was $\pm 7.9\%$, and on the Courbet
Peninsula, Îles Kerguelen females present fluctuated within $\pm 6.9\%$ between adjacent years,
and ± 9.85 and $\pm 7.89\%$ biannually¹⁹. Similarly, between 1995 and 1997, the population at
Peninsula Valdés reported interannual fluctuations of $\pm 5.93\%$ ⁴². These comparative records
support our assertion that the observed 46.9% decrease in females between 2022 and 2024 on
South Georgia is atypical.

One limitation of these counts is that they were not conducted on the same day across all
beaches in consecutive years. In 2024, counts were taken either on the same date (St Andrews
Bay) or one day later (Hound Bay and Gold Harbour) than in 2022 (Figure 1). However,
near-daily monitoring at St Andrews Bay and Hound Bay in 2024 showed a post-peak
attrition rate of -3.3% to -5.5% , accounting for only a small part of the observed decline.
Additionally, counts from the day before the corresponding 2022 dates were lower: 4,249 on
26/10/2024 (St Andrews Bay) and 1,112 on 25/10/2024 (Hound Bay). These findings
reinforce the atypical low signal in 2024, suggesting the one-day lag is unlikely to
significantly affect the observed decline.

Several plausible hypotheses exist that may explain components of the observed average
decline at South Georgia. One plausible contributor to the observed decline could be related
to the strenuous conditions of the HPAIV-impacted 2023 breeding season, which resulted in
numerous pup mortalities and/or abandonments (JC, Pers. Comm.). After losing their pup or
abandoning them due to their own HPAIV-induced stress, females may have left the breeding
beaches prematurely before oestrus, resulting in reduced copulation rates. Subsequently, this
would lead to fewer pregnancies and, ultimately, fewer females returning to give birth in the
following season. Limited evidence from Marion Island⁴³ suggests that at-sea copulation can
occur, potentially mitigating the impact of reduced on-land copulation rates on overall
reproductive success. However, these observations at Marion were limited to 2 individuals
over 15 cohorts, which may suggest that at sea-copulation does not occur at a magnitude

capable offsetting the observed decrease or that the two individuals observed at Marion
skipped the delayed implantation and bred during the moult instead. Consequently, further
follow-up monitoring in 2025 and beyond is needed to verify the ongoing impact of HPAIV
on recruitment.

Additionally, it is possible that, despite their strong philopatry^{44, 45}, some females may have
returned to other smaller, spatially removed outlier colonies or have skipped breeding and
simply returned to moult locations, which tracking data shows can differ from breeding
sites^{39, 46, 47}. Animals returning to outlier colonies would disperse the population more widely
along the coastline and reduce individual discrete counts at the larger beaches. However,
whilst shifts in populations have occurred as more suitable habitat emerges (i.e., the
emergence of the St Andrews Bay breeding site following the retreat of the Heaney and Cook
glaciers since the mid-1970s), such a dramatic shift deviates from typical behaviours of
aggregating at their preferred, discrete beaches.

A final possible factor is an external environmental influence, such as the unusual sea-ice
conditions observed in the South Atlantic during the austral winter of 2024. This anomaly
saw sea ice extending northward past the South Sandwich Islands and approximately 200km
south of mainland South Georgia, potentially influencing their distribution and foraging at
sea, which in turn may have hindered their post-breeding recovery from the 2023/24 season
(National Snow and Ice Data Centre - https://nsidc.org/data/seaice_index). However, given
the typically wide-ranging post-breeding behaviour of elephant seals⁴⁸, and their known
association with sea ice elsewhere in the Southern Ocean⁴⁹, the conditions in the South
Atlantic are unlikely to have had a significant impact on their post-breeding recovery or
triggered shifts in phenology.

While these hypotheses provide plausible explanations for some of the variation in population
numbers, they alone, or in combination, are unlikely to fully account for the dramatic decline
observed between 2022 and 2024. The temporal overlap of the arrival, and prevalence, of
HPAIV in the elephant seal population during this period coupled with the observed
reductions suggests a correlation that is too pronounced to be coincidental.

The long-term impact of the observed decline in the elephant seal population at South
Georgia is yet to be determined, underscoring the need for continued monitoring of this

globally important group. In Argentina, at Peninsula Valdés, HPAIV has had devastating
effects, with reported pup mortality of 70, 80 and 90% in in 2022, 2023 and 2024,
respectively; translating to approximately ~17,000 pup deaths³⁴. Due to the remoteness and
relative inaccessibility of breeding beaches around much of South Georgia, accurately
assessing pup mortality rates remains challenging. However, the rate of female absence from
the breeding beaches at South Georgia in 2024 exceeds those observed at Peninsula Valdés in
2023. Research on adjacent populations has shown that female survival is a critical
determinant of population growth^{50, 51}, and although we cannot be certain that all female
absences is due to mortality, it is probable that a significant portion of these absent seals have
perished. This loss of adult females is expected to adversely affect ongoing recruitment in the
immediate future, with the loss of multiple cohorts of pups impacting long term recruitment.
In addition, the final magnitude of HPAIV's impact on southern elephant seals is still
ongoing, with new reports of symptomatic animals occurring throughout the 2024/25 season
³³. It will not be until this virus mutates or a level of herd immunity is reached that the true
impact can be assessed.

Conclusion

The average 46.6% decline observed between 2022 and 2024 across South Georgia's three
largest breeding beaches is highly likely a direct consequence of HPAIV. This dramatic drop
contrasts sharply with historical interannual variations, which typically remain within 10%,
both locally and in comparable Southern Ocean populations. While external environmental
factors and behavioural shifts may have contributed, they alone cannot explain the severity of
this decline.

The scale of this event may have far-reaching consequences for the world's largest breeding
population of southern elephant seals at South Georgia. To assess its short-term impact and
determine whether the missing females represent true mortalities, follow-up monitoring in
2025 and 2026 is imperative. Additionally, to understand the long-term effects on
recruitment, more regular baseline population monitoring surveys need to be established to
monitor the major breeding beaches at regular intervals.

Data availability

All UAV survey data are available on request from the Polar Data Centre or from the
following DOIs: Data from the 2022/23 season for DPLUS109 are available for St Andrews
Bay at <https://doi.org/10.5285/f79b6577-d4cd-4eb2-9fec-b71e4e1d2389> &
<https://doi.org/10.5285/8189ed89-f36d-43c5-ae33-7e1c9ba0564d>; for Hound Bay at
<https://doi.org/10.5285/eb210711-85e1-4671-97a8-8c2791979a19>; and Gold Harbour at
<https://doi.org/10.5285/00cf7916-5c3f-42a2-8b98-38fa99b5aeb8>. Data for the 2024/25
season are available for St Andrews Bay at [https://doi.org/10.5285/e238f84e-63b2-4019-](https://doi.org/10.5285/e238f84e-63b2-4019-8308-7398a8ea204f)
[8308-7398a8ea204f](https://doi.org/10.5285/e238f84e-63b2-4019-8308-7398a8ea204f); for Hound Bay at [https://doi.org/10.5285/85de16ff-a3b2-42b4-a898-](https://doi.org/10.5285/85de16ff-a3b2-42b4-a898-aea94ee47b83)
[aea94ee47b83](https://doi.org/10.5285/85de16ff-a3b2-42b4-a898-aea94ee47b83); and for Gold Harbour at [https://doi.org/10.5285/47d0718d-7146-44d3-965c-](https://doi.org/10.5285/47d0718d-7146-44d3-965c-60e62a48b8cc)
[60e62a48b8cc](https://doi.org/10.5285/47d0718d-7146-44d3-965c-60e62a48b8cc).

Author contributions

Conceptualisation: CB, NF, JC & PH. Data collection: 2024: CB, NF, JC, CFC, JD; 2022:
NF, JC. Analysis: CB, JC & NF, with JC & NF leading in 2022. All authors contributed to the
development and final review of this manuscript.

Acknowledgements

This work was funded by the Biodiversity Challenge Fund Darwin Plus Main stage grants
DPLUS109 and DPLUS214. Fieldwork was conducted under Government of South Georgia
and the South Sandwich Islands Regulated Activity Permit Numbers 2022/021 and 2024/028,
and ASSI permits P/314 and P/444 & 445 for the 2022 and 2024 season, respectively. These
permits permitted BVLOS (Beyond Visual Line of Sight) UAV flights. Thanks also go to
colleagues at BAS and partners at GSGSSI for their comments on this manuscript. We would
also like to thank GSGSSI and the crew of MV Pharos SG for their logistical support. We
would also like to thank Lindblad National Geographic Expeditions and the crew of the NG
Explorer who kindly supported the field team with this work. Both fieldwork seasons
underwent review by the animal ethical approvals board at the British Antarctic Survey with
work approved and permitted under AWREB:1071 & 1109 in 2022 and 2024, respectively.

References

1. Ling JK, Bryden MM. Southern elephant seal — *Mirounga leonina*. In: *Handbook of Marine Mammals*
(eds Ridgway SH, Harrison RJ). Academic Press (1981).

2. Hindell MA, Perrin WF. Elephant Seals *Mirounga angustirostris* and *M. leonina*. In: *Encyclopedia of Marine Mammals* (eds Perrin WF, Wursig B, Thewissen JGM). 2nd edn. Academic Press (Elsevier) (2009).
 3. Boyd IL, Walker TR, Poncet J. Status of southern elephant seals at South Georgia. *Antarct Sci* **8**, 237-244 (1996).
 4. van Aarde RJ. Fluctuations in the population of southern elephant seals *Mirounga leonina* at Kerguelen Island. *Afr Zool* **15**, 99-106 (1980).
 5. Campagna C, Lewis M, Baldi R. BREEDING BIOLOGY OF SOUTHERN ELEPHANT SEALS IN PATAGONIA. *Mar Mamm Sci* **9**, 34-47 (1993).
 6. Le Boeuf BJ, Laws RM. *Elephant seals: An introduction to the genus*. University of California Press: Berkeley/Los Angeles, CA (1994).
 7. McCann TS. Population structure and social organization of Southern Elephant Seals, *Mirounga leonina* (L.). *Biol J Linn Soc* **14**, 133-150 (1980).
 8. Laws R. The Elephant Seal (*Mirounga leonina*, Linn.): II. General, social and reproductive behaviour. *Scientific Reports Falkland Islands Dependencies Survey* **13:88**, (1956).
 9. Carrick R, Csordas S, Ingham SE. Studies on the southern elephant seal, *Mirounga leonina* (L.). IV. Breeding and development. *CSIRO Wildlife Research* **7**, 161-197 (1962).
 10. Hindell MA, Burton HR. Seasonal Haul-Out Patterns of the Southern Elephant Seal (*Mirounga leonina* L.), at Macquarie Island. *J Mammal* **69**, 81-88 (1988).
 11. Slade RW, Moritz C, Hoelzel AR, Burton HR. Molecular Population Genetics of the Southern Elephant Seal *Mirounga leonina*. *Genetics* **149**, 1945-1957 (1998).
 12. Rus Hoelzel A, Campagna C, Arnbom T. Genetic and morphometric differentiation between island and mainland southern elephant seal populations. *Proc R Soc Lond, Ser B: Biol Sci* **268**, 325-332 (2001).
 13. Ferrari MA, Campagna C, Condit R, Lewis MN. The founding of a southern elephant seal colony. *Mar Mamm Sci* **29**, 407-423 (2013).
 14. Hindell MA, *et al*. Decadal changes in habitat characteristics influence population trajectories of southern elephant seals. *Global Change Biol* **23**, 5136-5150 (2017).
 15. McMahon CR, Bester MN, Hindell MA, Brook BW, Bradshaw CJA. Shifting trends: detecting environmentally mediated regulation in long-lived marine vertebrates using time-series data. *Oecologia* **159**, 69-82 (2009).
 16. Oosthuizen WC, Bester MN, Altwegg R, McIntyre T, de Bruyn PJN. Decomposing the variance in southern elephant seal weaning mass: partitioning environmental signals and maternal effects. *Ecosphere* **6**, art139 (2015).
 17. Barrat A, Mougín J. L'éléphant de mer *Mirounga leonina* de l'île de la Possession, archipel Crozet (46°25'S, 51°45'E). *Mammalia* **42**, 143-174 (1978).
 18. Guinet C, Jouventin P, Weimerskirch H. Population changes, movements of southern elephant seals on Crozet and Kerguelen Archipelagos in the last decades. *Polar Biol* **12**, 349-356 (1992).
 19. Guinet C, Jouventin P, Weimerskirch H. Recent population change of the southern elephant seal at Îles Crozet and Îles Kerguelen: the end of the decrease? *Antarct Sci* **11**, 193-197 (1999).
 20. Authier M, Delord K, Guinet C. Population trends of female Elephant Seals breeding on the Courbet Peninsula, îles Kerguelen. *Polar Biol* **34**, 319-328 (2011).

21. Laborie J, Authier M, Chaigne A, Delord K, Weimerskirch H, Guinet C. Estimation of total population size of southern elephant seals (*Mirounga leonina*) on Kerguelen and Crozet Archipelagos using very high-resolution satellite imagery. *Front Mar Sci* **10**, (2023).
 22. Wille M, Waldenström J. Weathering the Storm of High Pathogenicity Avian Influenza in Waterbirds. *Waterbirds* **46**, 100-109, 110 (2023).
 23. Adlhoch C, *et al.* Avian influenza overview February – May 2021. *EFSA Journal* **19**, e06951 (2021).
 24. Caliendo V, *et al.* Transatlantic spread of highly pathogenic avian influenza H5N1 by wild birds from Europe to North America in 2021. *Sci Rep* **12**, 11729 (2022).
 25. Alkie TN, *et al.* Recurring Trans-Atlantic Incursion of Clade 2.3.4.4b H5N1 Viruses by Long Distance Migratory Birds from Northern Europe to Canada in 2022/2023. *Viruses* **15**, 1836 (2023).
 26. Gamarra-Toledo V, *et al.* Avian flu threatens Neotropical birds. *Science* **379**, 246-246 (2023).
 27. Banyard AC, *et al.* Detection and spread of high pathogenicity avian influenza virus H5N1 in the Antarctic Region. *Nature Communications* **15**, 7433 (2024).
 28. Leguia M, *et al.* Highly pathogenic avian influenza A (H5N1) in marine mammals and seabirds in Peru. *Nature Communications* **14**, 5489 (2023).
 29. Ariyama N, *et al.* Highly Pathogenic Avian Influenza A(H5N1) Clade 2.3.4.4b Virus in Wild Birds, Chile. *Emerg Infect Dis* **29**, 1842-1845 (2023).
 30. Adlhoch C, *et al.* Avian influenza overview march–April 2023. *Efsa Journal* **21**, e08039 (2023).
 31. Lisovski S, *et al.* No evidence for highly pathogenic avian influenza virus H5N1 (clade 2.3.4.4b) in the Antarctic region during the austral summer 2022/23. *BioRxiv*, (2023).
 32. Bennison A, *et al.* A case study of highly pathogenic avian influenza (HPAI) H5N1 at Bird Island, South Georgia: the first documented outbreak in the subantarctic region. *Bird Study*, 1-12 (2024).
 33. SCAR. Sub-Antarctic and Antarctic Highly Pathogenic Avian INfluenza H5N1 Monitoring Project.) (2025).
 34. Campagna C, *et al.* Catastrophic mortality of southern elephant seals caused by H5N1 avian influenza. *Mar Mamm Sci* **40**, 322-325 (2024).
 35. Coleman J, *et al.* A comparison of established and digital surface model (DSM)-based methods to determine population estimates and densities for king penguin colonies, using fixed-wing drone and satellite imagery. *Remote Sensing in Ecology and Conservation* **n/a**, (2024).
 36. McCann TS, Rothery P. Population size and status of the southern elephant seal (*Mirounga leonina*) at South Georgia, 1951–1985. *Polar Biol* **8**, 305-309 (1988).
 37. Dickens J, *et al.* Developing UAV Monitoring of South Georgia and the South Sandwich Islands’ Iconic Land-Based Marine Predators. *Front Mar Sci* **8**, (2021).
 38. Fedak MA, Arnbom T, Boyd IL. The Relation between the Size of Southern Elephant Seal Mothers, the Growth of Their Pups, and the Use of Maternal Energy, Fat, and Protein during Lactation. *Physiol Zool* **69**, 887-911 (1996).
 39. McConnell BJ, Chambers C, Fedak MA. Foraging ecology of southern elephant seals in relation to the bathymetry and productivity of the Southern Ocean. *Antarct Sci* **4**, 393-398 (1992).
 40. Modig AO. Effects of body size and harem size on male reproductive behaviour in the southern elephant seal. *Anim Behav* **51**, 1295-1306 (1996).

41. Arnborn T, Fedak M, Boyd IL. Factors affecting maternal expenditure in southern elephant seals during
lactation. *Ecology* **78**, 471-483 (1997).
42. Lewis M, Campagna C, Quintana F, Falabella V. Estado actual y distribución de la población del
elefante marino del sur en la Península Valdés, Argentina. *Mastozoología Neotropical* **5**, 29-40 (1998).
43. de Bruyn PJN, Tosh CA, Bester MN, Cameron EZ, McIntyre T, Wilkinson IS. Sex at sea: alternative
mating system in an extremely polygynous mammal. *Anim Behav* **82**, 445-451 (2011).
44. Hindell MA, Little GJ. LONGEVITY, FERTILITY AND PHILOPATRY OF TWO FEMALE
SOUTHERN ELEPHANT SEALS (MIROUNGA LEONINA) AT MACQUARIE ISLAND. *Mar*
*Mamm Sci* **4**, 168-171 (1988).
45. Bester MN. MOVEMENTS OF SOUTHERN ELEPHANT SEALS AND SUBANTARCTIC FUR
SEALS IN RELATION TO MARION ISLAND. *Mar Mamm Sci* **5**, 257-265 (1989).
46. Tosh CA, *et al.* Adult male southern elephant seals from King George Island utilize the Weddell Sea.
*Antarct Sci* **21**, 113-121 (2009).
47. Bornemann H, *et al.* Southern elephant seal movements and Antarctic sea ice. *Antarct Sci* **12**, 3-15
(2000).
48. McConnell BJ, Chambers C, Fedak MA. Foraging ecology of southern elephant seals in relation to the
bathymetry and productivity of the Southern Ocean. *Antarct Sci* **4**, 393-398 (2004).
49. Labrousse S, *et al.* Coastal polynyas: Winter oases for subadult southern elephant seals in East
Antarctica. *Sci Rep* **8**, 3183 (2018).
50. De Bruyn P. Life history studies of the southern elephant seal population at Marion Island.). University
of Pretoria (2009).
51. Pistorius PA, Bester MN, Lewis MN, Taylor FE, Campagna C, Kirkman SP. Adult female survival,
population trend, and the implications of early primiparity in a capital breeder, the southern elephant
seal (*Mirounga leonina*). *J Zool* **263**, 107-119 (2004).

**Table 1 – Numbers of adult female Southern elephant seals present on the beaches during the breeding season.**
 Long term averages were calculated from available counts from 1958 to 2022. Records pre-1995 were obtained
 from ³⁶ and associated raw data from the BAS archives and represent both foot and boat-based counts of
 beaches. Counts from 1995 and 2019 were sourced from ^{3, 37}, respectively. Counts pre-1995 represent corrected
 counts to the peak of breeding assumed to be on the 25th October; counts from 1995 were made on the 17th and
 18th October for St Andrews Bay and Gold Harbour, respectively; counts from 2019 were both taken on the 25th
 October; and for 2022 and 2024, St Andrews Bay was counted on the 27th October, Gold Harbour on the
 10th November in both years, and Hound Bay on the 26th and 27th October in each year, respectively. NB¹: Counts
 from Gold Harbour in 2022 and 2024 were not taken close to the likely peak of breeding, and, consequently, are
 lower. NB²: At South Georgia, elephant seal sealing was conducted until the mid-1960s, and as such counts
 during this period represent a suppressed population.

	St Andrews Bay	Gold Harbour	Hound Bay	Combined average
1958	-	2611	-	
1959	-	2468	964	
1960	-	2243	-	
1961	-	2166	1270	
1963	-	1916	-	
1964	-	1833	2378	
1985	6198	4162	-	
1995	5719	3332	-	
2019	6074	-	2122	
2022	6305	1550	1901	
2024	4128	601	1066	
Long term average	6074	2476	1648	
SD	255	816	564	
% decrease from long term average	32	75.7	35.3	47.7 (SD = 24.3)
% decrease since 2022	34.5	61.2	43.9	46.6 (SD = 13.5)

506
507

St Andrews Bay

Hound Bay

Gold Harbour

2024

2022

17-Oct

18-Oct

19-Oct

20-Oct

21-Oct

22-Oct

23-Oct

24-Oct

25-Oct

26-Oct

27-Oct

28-Oct

29-Oct

30-Oct

31-Oct

01-Nov

10-Nov